# Effect of Deformation Parameters on Recrystallization Behavior and Long-Period Stacking-Ordered Phase of Mg-9Gd-4Y-2Zn-0.5Zr Alloy

**DOI:** 10.3390/ma15051822

**Published:** 2022-02-28

**Authors:** Rui Han, Jie Zheng, Zhaoming Yan, Leichen Jia, Jingjing Jia, Liang Liu, Zhimin Zhang, Yong Xue

**Affiliations:** 1School of Materials Science and Engineering, North University of China, Taiyuan 030051, China; hanrui19980914@163.com (R.H.); cqzhengjie@163.com (J.Z.); jlc226688@hotmail.com (L.J.); jiajingjing971121@163.com (J.J.); liuliang072921@163.com (L.L.); zhangzhimin@nuc.edu.cn (Z.Z.); 2Hubei Key Laboratory of Advanced Technology for Automotive Components, Wuhan 430070, China

**Keywords:** magnesium alloy, thermal compression, LPSO phase, dynamic recrystallization

## Abstract

In this study, a Mg-9Gd-4Y-2Zn-0.5Zr (wt.%) alloy was subjected, after solution treatment, to hot compression deformation at different temperatures (350 °C, 400 °C and 450 °C) and different strain rates (0.001 s^−1^, 0.01 s^−1^, 0.1 s^−1^ and 0.5 s^−1^) on a Gleeble-3800 thermal simulator. The evolution of the stress–strain curves under different conditions was compared. The changes in microstructure caused by the different deformation parameters and the change law of the long-period stacking-ordered (LPSO) phase during compression were observed and analyzed by optical microscope (OM) and scanning electron microscope (SEM). The results show that with the increase in the deformation temperature and the decrease in the strain rate, the degree of dynamic recrystallization (DRX) gradually increased, and the morphology of the phase also changed through, for example, twist fracture. The continuous dynamic recrystallization (CDRX) and discontinuous dynamic recrystallization (DDRX) mechanisms activated during the thermal deformation process can effectively refine the grains and weaken the texture in the alloy.

## 1. Introduction

Magnesium (Mg) alloys have been widely studied and applied in recent years due to their light weight, low density and high specific strength [1]. The addition of rare-earth elements has greatly improved the strength and machinability of Mg alloys due to solid-solution strengthening and precipitation strengthening, which has helped the application of rare-earth Mg alloys to spread widely [2]. The addition of Gd and Y can greatly improve the ductility and thermal stability of Mg alloys, which attenuates the shortcomings of the poor processing performance of Mg alloys at room temperature [3]. By adding a small amount of Zn into Mg-Gd-Y alloys, the LPSO phase of Mg-RE-Zn alloys can be generated [4]. The LPSO phase has the characteristics of high hardness, good thermal stability, good damping performance, high creep resistance and high elastic modulus [5]. Therefore, the effect of this new structural strengthening phase on the mechanical properties of Mg alloys has attracted extensive attention from researchers, and has become a heavy focus of research into the strengthening and toughening of Mg alloys [6].

The morphology, size and distribution of the LPSO phase can be controlled by different heat treatments and hot processing processes, thereby greatly improving the performance of Mg alloys [7]. In recent years, many studies have been devoted to observing the deformation behavior and precipitation dissolution of the LPSO phase during heat treatment or hot deformation. In a study of thermal processing, Wang et al. [8] studied the deformation behavior of Mg-Gd-Y-Zn-Zr alloy LPSO phase and its influence on grain size during cooling and forging-free deformation, and discussed the phenomenon of twist kink and the fragmentation of lamellar and block LPSO phases during deformation. Somekawa et al. [9] studied the effect of the volume fraction of long-period stacking-ordered (LPSO) phase on the high-temperature compression deformation behavior of Mg-Y-Zn alloys. The relationship between the volume fraction of the LPSO phase and the occurrence of deformation–torsion bands was understood and studied to further enhance the strength properties. Changes to the stress and strain in the deformation process can be accurately determined by the thermal compression process [10]. The forming ability of the material can be accurately judged by stress–strain curves and thermal processing maps [11]. The development of thermal compression technology helps to simplify the complex deformation behavior in the production process, explore its deformation theory and, in turn, guide the actual production. Therefore, it is necessary to the evolution of the phase and microstructure during hot compression deformation.

The hot deformation process involves the relationship between the flow stress and deformation conditions (strain rate and temperature), the deformation mechanism and the DRX behavior [12]. Recrystallization is another important aspect of thermal deformation behavior. In the thermal deformation process, DRX plays a soft role in the work hardening behavior and refines the grain, which can effectively improve the mechanical properties of the material. Li et al. [13] analyzed the effect of LPSO on the hot deformation and DRX behavior of a Mg-5.6Gd-0.8Zn alloy under different hot deformation parameters and explained the good thermal stability of the LPSO phase at low temperatures, the dissolution of the LPSO phase at high temperatures and the promotion of relative DRX behavior after dissolution. The good thermal stability of the LPSO phase at low temperatures, the dissolution of the LPSO phase at high temperatures and the promotion of relative DRX behavior after dissolution were explained. Prakash et al. [14] studied the effect of temperature on the hot deformation behavior of AZ80 alloys and mainly discussed the transformation mechanism of DRX from the CDRX to the DDRX of AZ80 alloys at different compression temperatures, as well as the existence of different morphological phases and their influence on the microstructure and texture evolution during deformation. However, the changes in the LPSO phase and microstructure of Mg-Gd-Y alloys under different thermal compression parameters are not perfectly known, especially for Mg-9Gd-4Y-2Zn-0.5Zr alloy, for which there is no complete research data support.

Therefore, in this study, a homogenized Mg-9Gd-4Y-2Zn-0.5Zr alloy was subjected to hot compression tests at different temperatures and different strain rates; the corresponding stress–strain curves were established according to the stress–strain values. The main changes in the microstructure and LPSO phase under different parameters were observed, and the influence of different temperatures and strain rates on the LPSO phase and recrystallization behavior was analyzed.

## 2. Materials and Methods

In this paper, Mg-9Gd-4Y-2Zn-0.5Zr alloy was studied. The as-cast billet was solution-treated at 520 °C for 24 h and then cooled in water [15]. Compared with as-cast alloy, the alloy after heat treatment had better performance. In addition, in order to better observe the phase transformation in the subsequent thermal deformation process, we chose the solid-solution treatment state, which can dissolve the as-cast phase into the matrix as the initial state. The temperature–time combination with the best performance and complete dissolution of the phase was obtained through experiments: 520 °C * 24 h. The solid-solution alloy was processed into 8 × 12 mm cylindrical specimens. The thermal compression tests at different temperatures and different strain rates were carried out on the specimens using Gleeble-3800 thermal torsion machine (DSI, Tucker, GA, USA) at 350 °C, 400 °C and 450 °C and strain rates of 0.001 s^−1^, 0.01 s^−1^, 0.1 s^−1^ and 0.5 s^−1^, respectively (selection of temperature and strain rate was close to the values used in industrial production). The deformation of the specimens was 60% (true strain: 0.91) and the heating rate was 10 °C/s. The specimens were preheated for 3 min before compression and cooled by water after compression. High-temperature-resistant lubricating oil and graphite sheets were uniformly coated on both ends of the sample to prevent the influence of friction and wear on the experimental results. After compression, the specimen was cut along the longitudinal section and characterized and analyzed. After sandpaper grinding and mechanical polishing, the corrosion was first carried out with the corrosion agents acetic acid, alcohol, water and picric acid; the microstructure was observed under an optical microscope (OM, Zeiss Axio Imager A2m, Oberochen, Germany). The surface-stress layer of the polished sample was taken out by ion thinning and placed under the electron microscope (SEM, Hitachi SU5000, Tokyo, Japan) in combination with the EDAX-TSL EBSD system (Hitachi SU5000, Tokyo, Japan) at 20 kV, 70° tilt angle and 15 mm working distance for EBSD test and SEM test. The EBSD data were analyzed by Channel 5 software (Hitachi SU5000, Tokyo, Japan), and the observation directions of all samples were parallel to the compression direction.

## 3. Results

### 3.1. Microstructure Analysis of Alloy before Deformation

Figure 1 shows the optical microscope microstructure and scanning electron microscope microstructure of the initial as-cast and solid-solution Mg-9Gd-4Y-2Zn-0.5Zr alloys. The microstructure of the as-cast Mg-9Gd-4Y-2Zn-0.5Zr alloy was mainly composed of large equiaxed grains and bone-like LPSO phase distributed at grain boundaries. The white part observed in the light microscope is the matrix; the black part is the LPSO phase. The LPSO phase observed in the scanning electron microscope image is bright white and the gray part is the matrix, which is consistent with other research results [16,17]. It can be observed from Figure 1b,d that the as-cast microstructure after solid-solution treatment underwent phase dissolution. The LPSO phase distributed at the grain boundary dissolved, while only the equiaxed grains with uniform distribution were left. The pinning effect during solution treatment effectively inhibited the grains’ growth. After homogenization, the grains grew slightly, and the average particle size was about 100 mm [18]. According to the comparison of the optical microscope images in Figure 1a,b, clear grain boundaries can be clearly seen in the alloy after solid solution. Comparing the scanning images before and after solid solution (Figure 1c,d), it was found that the dendritic LPSO phase distributed at the as-cast grain boundaries essentially dissolved completely and that the bright white part almost disappeared completely, with only the gray matrix part remaining [19].

### 3.2. Stress–Strain Curve Analysis

Figure 2 shows the stress–strain curves of the Mg-9Gd-4Y-2Zn-0.5Zr alloy during hot compression. Under various deformation conditions, with the increase in strain, the flow stress increased sharply at first. This was because with the increase in strain, the dislocation density increased significantly, which led to work hardening. This deformation stage is called the work-hardening stage; the duration of this stage is short. Subsequently, the growth rate of the stress slowed down. After reaching its peak, the flow gravity decreased slowly with the increase in strain. At this point, the work-hardening stage gradually changed to the deformation-softening stage, which was due to the occurrence of DRX in this process and the stress-softening phenomenon. After a long softening stage, the deformation-softening and work-hardening behavior caused by the DRX gradually tended to balance, while the flow stress fluctuation gradually decreased until it tended to be stable, thereby entering a stable stage. It is worth noting that under the condition of 350 °C/0.5 s^−1^, the stress–strain curve shows a strange trend: when the strain reached about 0.5 s^−1^, the stress decreased sharply, and then showed an upward trend. This may have been due to the low deformation temperature and high strain rate, which led to the kinking of the sample during the compression process. The grain orientation was less than the rotation in the direction that could produce slip, resulting in a reduced slip system and the inability to coordinate the deformation between grains [20].

By comparing (a), (b) and (c) in Figure 2, it can be observed that at the same deformation temperature, the overall stress decreased with the decrease in strain rate. When the strain rate was constant, with the increase in deformation temperature, the non-base slip system was activated, the dislocation climb and cross-slip activities increased and the stress decreased gradually. Therefore, in addition to the abnormal curve caused by the kink at low temperature and high strain rate, the maximum stress value of 243 MPa was generated at a low temperature and high strain rate, while the minimum stress value of 48 MPa was generated at a high temperature and low strain rate.

### 3.3. Microstructure Changes at Different Strain Rates

The microstructure at different strain rates at 350 °C was observed and analyzed. The sampling position was the central deformation area. The different colors in the IPF diagram represent the different orientation angles of the Mg alloy grains. The black part in the IPF diagram is the phase that cannot be identified in rare-earth Mg. The large grains in all the states were compressed along the compression direction (CD) and elongated in the vertical direction. The massive LPSO phase at the grain boundary was precipitated and accompanied by the grain boundary deformation of the large grains, which were compressed into strips and then broken. Due to the occurrence of DRX during the compression, the microstructure of the alloy exhibited a bimodal microstructure, but the DRX was different at different strain rates [21,22]. With the decrease in the strain rate, the degree of recrystallization gradually increased. Figure 3a–c is the microstructure of the alloy at a strain rate of 0.5 s^−1^. It can be seen from the figure that at 350 °C/0.5 s^−1^, the microstructure of the alloy was uneven, DRX basically did not occur and the original equiaxed grain and phase change were not obvious. This may be because the low temperature and relatively high strain rate led to the kinking of the alloy sample, while the microstructure deformation was extremely irregular and DRX behavior and grain boundary slip rarely occurred, which is consistent with the abnormal changes in the curve in Figure 2. The phenomenon shown in the microstructure is also consistent with the previous speculation. With the decrease in strain rate, the recrystallization behavior occurred at strain rates of 0.1 s^−1^ and 0.01 s^−1^ and, as indicated by the yellow arrow in Figure 3d, the recrystallization began at the grain boundary with higher stress. When the strain rate was reduced to 0.001 s^−1^, obvious recrystallization grains appeared at the grain boundary, the large deformed grains were surrounded by recrystallized grains, and the bimodal structure was obvious. The scanning diagram in Figure 3k also clearly demonstrates the recrystallization microstructure. The grain size was also reduced from 68.91 µm at a strain rate of 0.5 s^−1^ to 21.88 µm. With the decrease in strain rate, it can be seen that the block LPSO phase in the scanning diagram also changed. With the decrease in strain rate, the LPSO phase gradually changed from the initial large dendrite to a long strip. With the continuous decrease in strain rate, it can be seen that the long-strip LPSO phase was broken into intermittent short strips or broken into small pieces.

### 3.4. Microstructure Changes at Different Deformation Temperatures

Figure 3 shows that the lower the strain rate, the higher the DRX degree and the longer the phases took to break. Therefore, a 0.001 s^−1^ strain rate was selected to analyze the microstructure at different temperatures, as shown in Figure 4. When deformed at 400 °C, the degree of recrystallization at 350 °C/0.001 s^−1^ in Figure 3 was improved, but there were still large grains, while small grains were generated around the boundaries of the large grains, forming an obvious bimodal microstructure. The grain size decreased from 21.888 to 19.54. With the increase in temperature to 450 °C, it can be observed that the degree of crystallization significantly improved, indicating that recrystallization was more likely to occur with the increase in temperature. At 450 °C/0.001 s^−1^, most large grains were transformed into many DRX grains.

### 3.5. The Evolution of the LPSO Phase

In order to further determine the evolution of the LPSO phase under different deformation conditions, the black dotted frame was amplified at high magnification, as shown in Figure 3b,h and Figure 4b,e, respectively. During the deformation process, the LPSO phase exhibited a large branch shape similar to the as-cast state due to the unstable deformation conditions at the low temperature and high strain rate; as shown in Figure 5a, the edge was accompanied by lamellar trailing. No obvious precipitation of granular phase was found in the phase shown in the diagram [23]. When the compression strain rate was reduced to 0.01 s^−1^ at 350 °C, the LPSO phase was compressed into a slender strip. It can be seen that the lamellar trailing was still present, but there were different degrees of twist. The twist deformation of LPSO phase changes the direction of the dislocation movement, prevents dislocation accumulation at grain boundaries, and cannot reach the energy required by DRX [24]. When the deformation temperature was 400 °C and the strain was 0.001 s^−1^, the LPSO phase was still in the form of a slender strip, as shown in Figure 5c. However, due to the increase in temperature and the decrease in strain rate, large energy accumulation occurred at the grain boundary. With the occurrence of recrystallization behavior, the LPSO phase also broke and dissolved, forming a short strip and a small lamellar area. As the temperature continued to rise to 450 °C, the DRX was close to complete at a low strain rate. At 450 °C × 0.001 s^−1^, the LPSO phase was broken into small pieces, the LPSO phase was completely broken into small pieces and the volume was also significantly reduced compared with the 350 °C/0.5 s^−1^ state shown in Figure 5a.

## 4. Discussion

It is known that the grain refinement in the deformation process mainly comes from DRX. In order to further analyze the DRX mechanism of the alloy in the thermal deformation process, the grain size (the position of the black dotted frame in Figure 4f) was selected from Figure 4f for the amplification analysis, as shown in Figure 6a. The point-to-point misdirection and point-to-origin misdirection of arrows from A to B and from C to D, respectively, are shown in Figure 6b, c [25]. In the deformation process, the lattice inside the grain rotated, resulting in point-to-point directional distortion. In Figure 6b, the orientation deviation of the point-to-origin shows an overall increasing trend, gradually increasing to 12.57. Many low-angle grain boundaries (LAGBs) (white solid lines referred to by the yellow arrow in the figure) were distributed in the large grains, which were usually formed by dislocation accumulation and subgrain boundaries. During the deformation process, the dislocation density increased gradually and dislocation migration occurred. The LGABs continuously captured dislocations, resulting in the transformation of the LAGBs into high-angle grain boundaries (HAGBs) (marked by the black solid lines referred to by the green arrow in the figure) and the formation of new DRXed grains in the large grains. In Figure 6c, the curve value suddenly rises, which shows a CDRX process. The grain boundaries around the small grains were jagged, and part of the grain boundaries uplifted to the adjacent grains. Subgrains formed at the black grain boundaries at the grain edges and were gradually separated by the LAGBs from the coarse grains to form DRXed grains, showing a typical DDRX process. This indicates that both the CDRX mechanism and the DDRX mechanism were present during the hot deformation. The evolution of this mechanism has been reported in magnesium alloys prepared by extrusion and other methods [26,27].

In Figure 7, the coarse deformed grains and dynamic recrystallized grains at a high temperature and a low strain rate of 450 °C/0.001 s^−1^ are presented (the DRXed grains in this study were defined as grains with a grain orientation spread (GOS) of less than 2°) [28]. By comparing the pole figures of the deformed grains (Figure 7b) and recrystallized grains (Figure 7d), it can be seen that the coarse deformed grains had strong basal texture strength; the maximum value was 13.677. The texture strength distribution of the recrystallized grains was similar to that of the large grains, but the texture strength was only 3.378, which proves that DRX can significantly weaken the texture strength of grains.

## 5. Conclusions

With the increase in deformation temperature and the decrease in strain rate, the degree of recrystallization gradually increased, so the degree of recrystallization was largest at 450 °C/0.001 s^−1^ in the experimental range.The changes in the deformation conditions during compression resulted in precipitation, dissolution and kinking of the LPSO phase. At 350 °C/0.5 s^−1^, the LPSO phase presented a dendritic morphology similar to that of the as-cast state. At 350/0.01 s^−1^, the twist and breakage of the LPSO phase were generated. At 450 °C/0.001 s^−1^, the LPSO phase was broken into small pieces.The CDRX and DDRX mechanisms were the dominant deformation behaviors during the thermal compression, which contributed to texture weakening and grain refinement.

## Figures and Tables

**Figure 1 materials-15-01822-f001:**
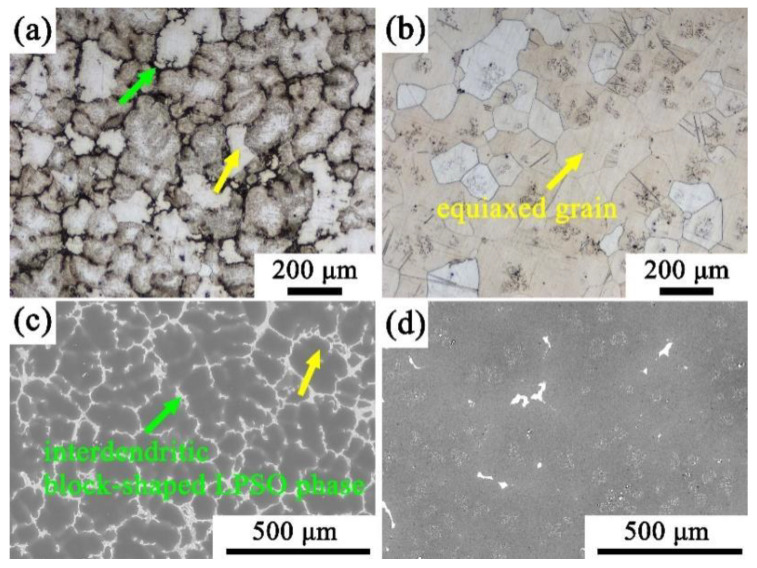
Microstructure of as-cast (**a**,**c**) and homogenized (**b**,**d**) alloys ((**a**,**b**) OM, (**c**,**d**) SEM).

**Figure 2 materials-15-01822-f002:**
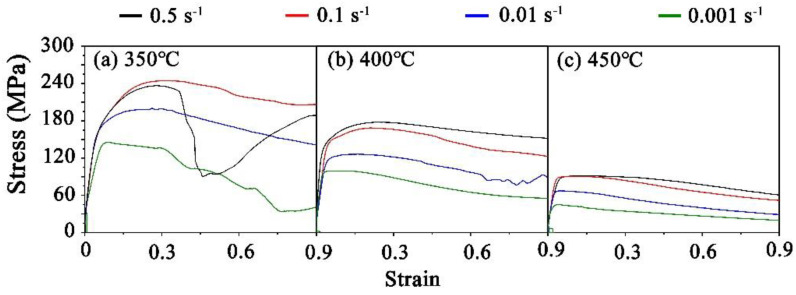
Stress–strain curves under different parameters: (**a**) 350 °C (**b**) 400 °C (**c**) 450 °C.

**Figure 3 materials-15-01822-f003:**
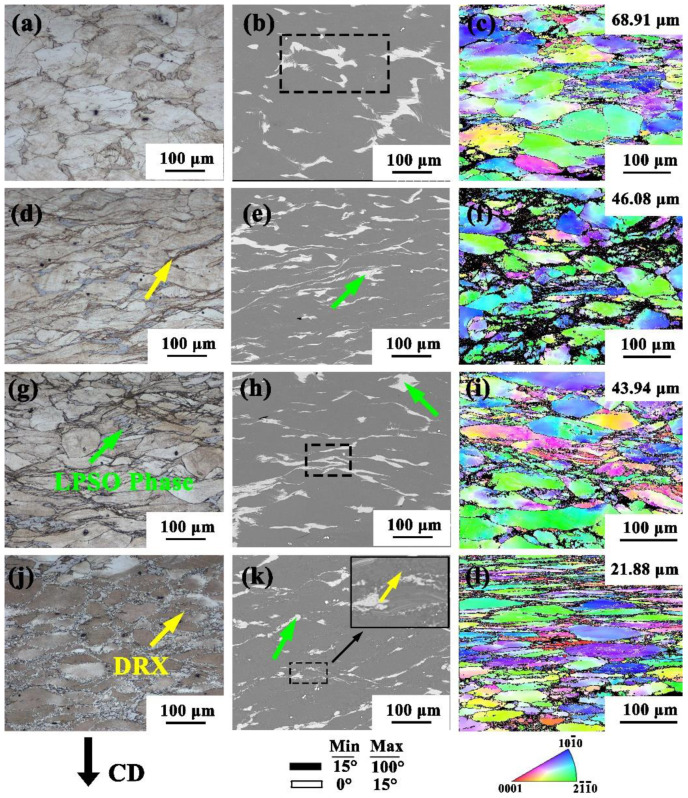
OM (**a**,**d**,**g**,**j**), SEM (**b**,**e**,**h**,**k**), EBSD (**c**,**f**,**i**,**l**) at strain rate from high to low (0.5 s^−1^, 0.1 s^−1^, 0.01 s^−1^, 0.001 s^−1^) at 350 °C.

**Figure 4 materials-15-01822-f004:**
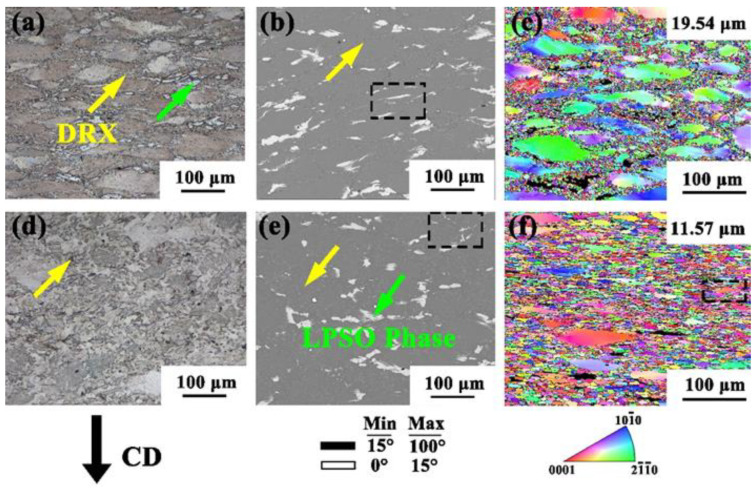
OM (**a**,**d**), SEM (**b**,**e**) and EBSD (**c**,**f**) at strain rates of 0.001 s^−1^ and temperatures of 400 °C (**a**–**c**) and 450 °C (**d**–**f**).

**Figure 5 materials-15-01822-f005:**
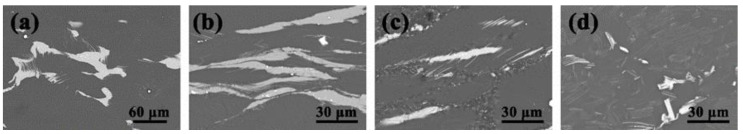
High-magnification SEM images under different deformation conditions. (**a**) 350 °C/0.5 s^−1^ (**b**) 350 °C/0.01 s^−1^ (**c**) 400 °C/0.001 s^−1^ (**d**) 450 °C/0.001 s^−1^.

**Figure 6 materials-15-01822-f006:**
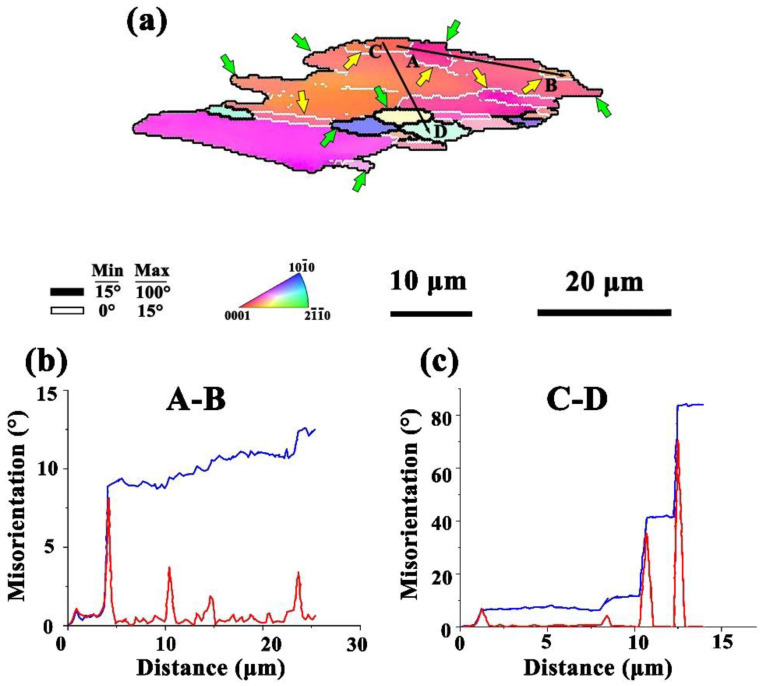
(**a**) Extract grains from the sample in Figure 4f (highlighted in the dotted frame). Typical point-to-point and point-to-origin orientation errors along the arrows marked in (**a**), from A to B (**b**) and from C to D (**c**).

**Figure 7 materials-15-01822-f007:**
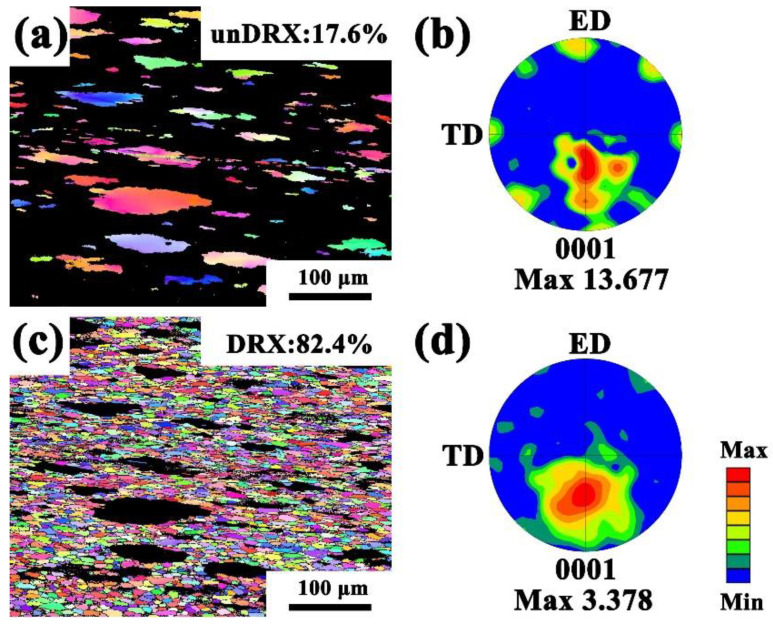
450 °C/0.001 s^−1^ EBSD results of deformed grains and dynamic recrystallized grains (**a**,**b**) of the sample (**c**,**d**) DRXed grains (**a**,**c**) IPF maps (**b**,**d**) (0001) pole figures.

## Data Availability

Data are contained within the article.

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
