# Peer review of "Effect of Deformation Parameters on Recrystallization Behavior and Long-Period Stacking-Ordered Phase of Mg-9Gd-4Y-2Zn-0.5Zr Alloy"

_materials, 2022, doi:10.3390/ma15051822_

Round 1

Reviewer 1 Report

This is a very interesting paper regarding the investigation one of magnesium alloy (Mg-9Gd-4Y-2Zn-0.5Zr alloy). Magnesium alloy are in the focus of different investigations in recent years due to the (light weight, low density and high specific strength). The title of the paper correspond well with the content. The abstract shows a concise summary of the paper. The research study methods are sound and accurate. The conclusions are accurate and supported by the content. The literature review and research study methods are explained clearly.  I have only a few suggestions to the authors:

Please do not use the abbreviation in the title (LPSO) without its explanation (like you do for abbreviation in the abstract section)

Please give us some more detail about the mechanical preparation of specimen 

Please give us details about optical microscope (model, manufacturer).

The designation of the SEM should be Hitachi Field Emission Scanning Electron Microscope SU5000.

Furthermore, I have also check paper for the plagiarism with Turnitin program which I also attach. Although the total overlap percentage is relatively large, this is due to the large number of sources, where the overlap is up to 1% and the largest single overlap is 3%. By reviewing the text, I do not find any traces of plagiarism in the paper and for that reason I think that no corrections should be made on this issue. 

Reviewer 2 Report

Your work makes a good impression, especially when it comes to the analysis of changes in the microstructure of the alloy after deformation.
In subsequent works, it would be possible to show the dynamics of the change in the dislocation density near the grain boundaries.

Reviewer 3 Report

This manuscript reports an experimental research on the hot compression behaviour of a polycrystalline Mg alloy.  The stress-strain tests were performed under three different temperatures and four deformation rates. The effects of changing these parameters on microstructure were investigated, highlighting the dominant mechanisms during hot deformation that contribute to the final microstructure.

The alloy considered (Mg-9Gd-4Y-2Zn-0.5Zr) or similar has been the subject of many investigations in recent years. According to the authors, the novelty of their paper consists of the description of changes of the LPSO phase and the alloy microstructure during compression under three different temperatures and four strain rates. The work deserves to be published, however I have the following questions.

  • Line 74 What do you mean for “perfect”? Perhaps “perfectly known”?
  • Line 84 Since mechanical behaviour and microstructural changes depends on the initial metallurgical state, I ask the authors to better clarify the reason for the choice of this initial heat treatment.
  • Line 89 Please, also give the respective values of the homologous temperature.
  • Line 163-209 Concerning the result presentation, I would like a description of the statistical methodology for grain size determination to be shown. In particular, how was the average grain size calculated?
  • Line 200-209 I wonder if the degree of recrystallization was quantitatively defined or only qualitatively inferred.
  • I would like to know if you verified the absence/presence of strengthening nanoparticles precipitation during dynamic recrystallization, which was reported in literature for a similar alloy (Huan Liu 2017 https://doi.org/10.3390/met7100398 )
  • Finally, it would be interesting to compare the results of this work, which concern dynamic recrystallization, with those of a previous paper of the same authors ( Rui Han et al. 2021 https://doi.org/10.1088/2053-1591/ac39c1  ) which deals with static recrystallization.

I think that with these clarifications the work will be more complete and attractive for potential readers. My general recommendation is in favour of publishing the manuscript after the required revision.
